# Effect of *Corynebacterium glutamicum* Fermentation on the Volatile Flavors of the Enzymatic Hydrolysate of Soybean Protein Isolate

**DOI:** 10.3390/foods13162591

**Published:** 2024-08-19

**Authors:** Lingling Shangguan, Zixiong Liu, Linglong Xu, Qiao Yang, Xiaoling Zhang, Lan Yao, Pei Li, Xiong Chen, Jun Dai

**Affiliations:** 1Key Laboratory of Fermentation Engineering (Ministry of Education), Cooperative Innovation Center of Industrial Fermentation (Ministry of Education & Hubei Province), National “111” Center for Cellular Regulation and Molecular Pharmaceutics, Hubei Key Laboratory of Industrial Microbiology, School of Life Sciences and Health, Hubei University of Technology, Wuhan 430068, China; shangguanlingling@hbut.edu.cn (L.S.); liuzixiong@hbut.edu.cn (Z.L.); xulinglong7149@163.com (L.X.); yaolislan1982@aliyun.com (L.Y.); 2ABI Group, Laboratory of Phycosphere Microbiology, College of Marine Science and Technology, Zhejiang Ocean University, Zhoushan 316022, China; qiaoyang1979@whu.edu.cn (Q.Y.); zhangxiaoling@zjou.edu.cn (X.Z.); 3Hubei Key Laboratory of Yeast Function, Angel Yeast Co., Ltd., Yichang 443000, China; lipei4041@126.com

**Keywords:** volatile flavoring compound, soybean protein isolate, *Corynebacterium glutamicum*, fermentation, seasoning

## Abstract

This study focused on improving the flavor quality of seasonings, and enzymatic hydrolysis of soybean protein isolate (SPI) seasoning via traditional technology may lead to undesirable flavors. Herein, we aimed to develop a new type of SPI seasoning through microbial fermentation to improve its flavor quality. The effect of *Corynebacterium glutamicum* fermentation on the flavoring compounds of seasonings in SPI enzymatic hydrolysate was examined. Sensory evaluation showed that the SPI seasoning had mainly aromatic and roasted flavor, and the response signals of S18 (aromatic compounds), S24 (alcohols and aldehydes), and S25 (esters and ketones) sensors of the electronic nose differed significantly. Overall, 91 volatile compounds were identified via gas chromatography–mass spectrometry. SPI seasonings contained a higher number of alcohols, ketones, aromatics, and heterocyclic compounds than traditional seasonings, which had stronger cheese, fatty, and roasted aromas. According to the relative odor activity value (*ROAV*) analysis, n-pentylpyrzine, 2,6-dimethylpyrazine, and tetramethylpyrazine are the key flavoring compounds (*ROAV* ≥ 1) of SPI seasoning, which may impart a unique roasted and meaty aroma. Therefore, the fermentation of SPI enzymatic hydrolysate with *C. glutamicum* may improve the flavor quality of its products, providing a new method for the development and production of new seasoning products.

## 1. Introduction

Good flavor is a key factor that affects food quality, which is the combination of taste, smell, and mouthfeel, and is usually related to specific volatile flavoring compounds such as aldehydes, alcohols, ketones, terpenes, and aromatic compounds [1]. These compounds contribute to food sweetness and aroma, improve undesirable flavor, and alleviate the loss of appetite. Hence, food flavor has become a hot topic in current research. Soybean protein isolate (SPI) is a plant food raw material that contains high levels of proteins, nutrients, aliphatic compounds, aromatic compounds, heterocyclic compounds, terpenes, and other key flavoring compounds [2]. SPI enzymatic hydrolysate is obtained from the enzymatic hydrolysis of soy protein isolate. As a flavor precursor, SPI is mainly used in the research of seasoning base material [3]. SPI foods prepared via commercial proteases often have unpleasant odors such as bitter taste and bean odor, which directly affect the sensory properties of SPI enzymatic hydrolysate [4]. Therefore, the flavor profile of SPI seasoning must be improved for its further application and development.

Microbial fermentation is considered a sustainable method for excellent flavor formation. This method effectively improved the sensory quality and overall characteristics of bean products [5]. For example, Liang et al. [6] enriched volatile flavoring compounds of mung bean powder through lactic acid bacteria fermentation, which reduced unpleasant flavor and improved the flavor and functional characteristics of the mung bean powder. Bai et al. [7] also examined the effect of *Lactiplantibacillus plantarum* and *Saccharomyces cerevisiae* fermentation on the volatile flavoring compounds of highland barley and revealed significant changes in these compounds; moreover, the flavor and functional characteristics of highland barley were effectively improved. However, the formation of good flavoring compounds is dependent on the selection of microorganisms and ingredients. SPI enzymatic hydrolysate is considered an essential substrate for the promotion of microbial growth and accumulation of volatile flavoring compounds [8]. Therefore, to overcome the problem of unpleasant flavor, this study prepared a new type of seasoning via microbial fermentation of SPI enzymatic hydrolysate to improve the characteristics of flavoring compounds.

*C. glutamicum* is an ideal microbial chassis for industrial production, with advantages such as biosafety, low nutrient requirements, rapid growth, and broad substrate spectrum [9]. Previously, we revealed that this strain can industrially produce the flavor enhancer L–glutamate from carbohydrates, which imparts a unique flavor and texture to fermented products [10]. The use of SPI enzymatic hydrolysate as the main raw material of the culture medium may significantly improve the properties of various umami substances of fermented products, and *C. glutamicum* is a potential fermentation strain for the industrial production of new seasonings with improved flavoring compounds [11]. Only a few studies have reported the use of *C. glutamicum* to produce economical and safe flavoring substances from enzymatic soybean meal hydrolysates and confirmed that it may produce other excellent flavoring substances [12]. Therefore, to prepare a new seasoning, *C. glutamicum* could be used for the fermentation of SPI enzymatic hydrolysate to enrich its volatile flavoring compounds and improve its flavor quality.

To analyze and characterize the overall flavor and subtle differences in volatile flavoring compounds of different seasoning samples, quantitative descriptive analysis (QDA), electronic nose (E–nose) technology, and gas chromatography–mass spectrometry (GC–MS) were employed. QDA, which is a highly intuitive and comprehensive sensory evaluation method, is widely used for the systematic analysis of food flavor profiles. E-nose serves as an efficient, time-saving, and low-cost method for detecting the overall odor of a seasoning sample, providing a more accurate and objective indication of the sample’s quality than traditional sensory evaluations [13]. GC–MS is a compound-type detection technology with a high resolution and accuracy that can be used to detect volatile and nonvolatile compounds in hot sauce and other seasoning samples [14]. Compared with GC–ion migration spectroscopy (GC–IMS), which only focuses on the detection of volatile flavoring compounds, GC–MS has a wider detection range. All components of unknown compounds in the sample can be quantitatively analyzed [15]. Therefore, GC–MS can be used to comprehensively analyze the flavor profiles of seasonings.

In this study, QDA and E-nose were used to identify the overall flavor profiles of different seasoning samples, and the combination of GC–MS with statistical multivariate analysis tools (principal component analysis [PCA]) was used to systematically characterize and distinguish volatile flavoring compounds in different seasoning samples. In addition, key flavor active components were identified based on their relative odor activity value (*ROAV*), and the correlation between sensory properties and important flavoring compounds was investigated using PCA. This research is beneficial in enhancing the added value of SPI and providing a theoretical basis for developing new seasonings.

## 2. Materials and Methods

### 2.1. Materials and Chemicals

Yeast powder was purchased from Angel Yeast Co., Ltd. (Yichang, China), SPI (food grade) from Linyi Shansong Biological Products Co., Ltd. (Linyi, China), L-methionine (purity ≥ 99%) and biotin (purity ≥ 99%) from Guangzhou Saiguo Biotechnology Co., Ltd. (Guangzhou, China), vitamin B1 (purity ≥ 99%) from Shanghai Yuanye Biotechnology Co., Ltd. (Shanghai, China), and other chemicals (analytically pure) from Sinopharm Chemical Reagent Co., Ltd. (Beijing, China).

*Corynebacterium glutamicum* P-45 was selected as the fermentation strain, which was obtained via atmospheric and room-temperature plasma mutagenesis [10]. The fermentation medium of this strain contained glucose (27.0 g/L), yeast powder (41.0 g/L), succinic acid (1.0 g/L), urea (12.0 g/L), MgSO_4_·7H_2_O (0.4 g/L), L-methionine (0.5 g/L), K_2_HPO_4_·3H_2_O (2.4 g/L), biotin (0.3 mg/L), vitamin B1 (0.2 mg/L), and MnSO_4_·H_2_O (10.0 mg/L).

### 2.2. Preparation of Three Seasoning Samples

In this study, sample S was prepared via the traditional enzymatic hydrolysis of SPI, sample F was prepared via the traditional *C. glutamicum* fermentation method, and sample SF was prepared via the *C. glutamicum* fermentation method using SPI enzymatic hydrolysate. The preparation steps of the three samples were as follows:

S was prepared according to the enzymatic hydrolysis process optimized in our previous experiments (20,000 U/g neutral protease and complex protease were added to soy protein isolate in a 1:1 ratio under the following conditions: pH 7.0, substrate concentration of 15 g/100 mL, and hydrolysis at 45 °C for 13 h) [16]. F was prepared via *C. glutamicum* fermentation according to the parameters obtained in previous experiments [11]. SF was prepared by adding 40% (*v*/*v*) SPI enzymatic hydrolysate into the fermentation medium as a substrate for *C. glutamicum* and using the fermentation parameters of F [11].

The abovementioned samples were prepared in triplicate. After centrifugation (12,000 rpm for 10 min), the supernatants obtained from the seasoning samples prepared via different methods were tested. Each sample was tested thrice, and the average was considered the final effective value.

### 2.3. Sensory Evaluation

QDA was used for sensory analysis [17]. The sensory evaluation team comprised 12 students majoring in food (6 males and 6 females, aged 20–30 years), who received training in professional sensory assessment to evaluate the odor characteristics in terms of woody, sour, beany, grassy, fatty, and aroma flavors. The strength of each property was recorded from 1 (low strength) to 8 (high strength). The sensory experiments were conducted, including the smelling of seasoning samples obtained via fermentation with *C. glutamicum*, a generally recognized as safe microbe [18], in accordance with national regulations (China). Moreover, the medium used in the process was safe and harmless. Therefore, no further ethical permission was required. Appropriate protocols for protecting the rights and privacy of all participants were utilized during the execution of this research. Informed consent was obtained from all participants.

### 2.4. E-Nose Measurement

cNose-18 electronic nose (Shanghai Baosheng Industrial Development Co., Ltd., Shanghai, China) was used to analyze the overall odor characteristics of the seasoning samples [19]. The E-nose system was equipped with 27 sensors that have varying degrees of response to carbonaceous substances and nitrogenous substances; alkanes; and aromatic compounds such as alcohols, aldehydes, and ketones (Table A1). Moreover, 30 mL of the seasoning sample was diluted to 60 mL in a 1:1 (water-to-seasoning sample) ratio. The diluted seasoning sample was then evenly placed into a special bottle for E-nose analysis. Before testing, each seasoning sample was sealed with a plastic wrap and maintained at 37 °C for 30 min. The detection and cleaning times were set to 120 and 100 s, respectively. Different seasoning samples were measured simultaneously eight times. To reduce systematic errors, the stable values of the last four measurements were taken as response values for analysis [20].

### 2.5. GC–MS Analysis

Volatile flavoring compounds present in seasoning samples were detected using HP6890/5975C GC–MS (Agilent, CA, USA).

Sample treatment: To extract volatile compounds, each sample (5 g) was mixed with 3 mL of saturated NaCl and 1 μL of the internal standard (1,2-dichlorobenzene; 221 μg/mL) in a 20 mL headspace extract bottle. With a magnetic stirrer (Gongyi Yuhua instrument Co., Ltd., Gongyi, China), the mixture is thoroughly mixed for 1min at a speed of 1300 rpm to disperse the sample and ensure that the internal standard was evenly mixed in the sample. The headspace bottle containing the mixture was then placed in a 50 °C water bath and oscillated at 300 rpm for 30 min before the headspace sampling to balance its headspace. Then, the SPME (Supelco, Bellefone, PA, USA) fibers, which had been treated at 250 °C for 60 min in the gas chromatograph inlet, were exposed to the top space of the headspace vial and extracted at 50 °C for 20 min. Once the headspace volatiles had been collected, the fibers were retrieved and immediately transferred to the GC instrument’s injector. The sample purge time, sample preheating time, sample desorption temperature, and sample purge temperature were 30 min, 5 min, 180 °C, and 70 °C, respectively. The seasoning samples were tested for volatile flavoring compounds under specific GC and MS conditions.

GC conditions: Chromatography was performed using an Agilent DB-WAX column (30 m × 0.25 mm × 0.25 μm) at a column temperature of 40 °C (hold for 2 min). The column was heated up to 250 °C at a rate of 2 °C/min for 5 min. The total running time was 112 min, and the vaporization chamber temperature was 250 °C. The carrier gas was high purity He (99.999%), and the precolumn pressure, carrier gas flow rate, and solvent delay time were 18.411 psi, 3.0 mL/min, and 2 min, respectively [21].

MS conditions: The ion source, ion source temperature, quaternary bar temperature, electron source energy, and activation voltage were Electron impact ion source (EI), 230 °C, 150 °C, 70 eV, and 1.6 V, respectively [22]. Data analysis was performed using Qualitative Navigator B.08.00 software.

Volatile compounds were identified by comparing their linear retention indexes (RIs) with the linear RI of the alkane standard solution (C7–C30) based on a comparison of the mass spectrum with the available mass spectrum in the NIST 11 mass spectrum library. Samples with a matching degree of >80% were extracted and analyzed. Based on the method of Wang et al. [23], the relative percent content of each volatile compound was calculated by dividing the peak area of a single compound by that of the total compound. All analyses were performed in triplicate.

### 2.6. ROAV

*ROAV* can be used to assess the contribution of each volatile flavoring compound to the overall aroma and flavor of the seasoning sample [24]. Owing to the variety of volatile flavoring compounds in seasoning samples (such as alcohols, aldehydes, ketones, esters, acids, aromatics, and heterocycles), these samples often contain hundreds of volatile flavoring compounds. To more conveniently analyze the influence of various volatile flavor substances on the overall aroma and flavor of the seasoning sample, *ROAV* was used to evaluate the contribution of each volatile flavoring compound [25], and the compound that contributed the most to the odor of the seasoning sample was defined as *ROAVmax*, which was given a value of 100. The *ROAV* of other volatile flavoring compounds was calculated as follows:$${ROVA}_{i}\approx 100\times \frac{C_{i}}{C_{\max}}\times \frac{T_{\max}}{T_{i}}$$
where *C_i_* and *T_i_* represent the relative content (%) (the peak intensity of each compound to the total peak intensity of all compounds) and sensory threshold (mg/kg) of each volatile flavoring compound, respectively. *C_max_* and *T_max_* represent the relative content (%) and sensory threshold (mg/kg) of the compound that contributed the most to the total odor of the seasoning sample, respectively. The threshold for *ROAV* calculation was derived from the olfactory threshold in water obtained from the literature [26].

In this method, *ROAV* < 1 indicates that the substance has a lower contribution to the overall flavor of the product, whereas *ROAV* > 1 indicates that the substance has a greater contribution to the overall flavor of the product.

### 2.7. Statistical Analysis

All experiments were repeated thrice, and the experimental results were expressed as mean ± standard deviation. IBM SPSS Statistics for Windows version 23 (IBM Corp., Armonk, NY, USA) was used for analysis, and the Duncan multiple range test was used to assess significance; *p*-values of <0.05 were considered to indicate statistical significance. Charts were constructed using OriginPro 2021, and PCA (PCA-X) and orthogonal partial least squares discriminant analysis (OPLS-DA) were performed via SIMCA 14.1.

## 3. Results and Discussion

### 3.1. Sensory Evaluation of Seasoning Samples

To identify the flavor profiles of different seasonings, the flavor profiles of three seasonings were evaluated via sensory scoring (Figure 1a). In general, the sensory properties of fermentation seasonings are affected by raw materials, preparation technology, and other conditions. The results showed that the acidic odor of samples S, F, and SF was not noticeable, which may be due to the high odor threshold of acid; however, this had little effect on the overall sour taste of the seasoning samples [27]. Samples F and SF mainly had aromatic and roasted flavor. Compared with F, the aromatic and roasted flavor of SF were significantly increased (*p* < 0.05), which may be because SPI samples produce more volatile flavoring substances such as ketones, aromas, and heterocycles during fermentation [7]. The bean odor score was significantly higher in S (*p* < 0.05) than in F and SF, which may be attributed to the presence of some odorous compounds produced during enzymatic hydrolysis [28]. In addition, sample S showed low scores in almost all sensory attributes, except for beaniness, which may be due to the relatively small number or type of compounds detected in the seasoning sample.

### 3.2. E-Nose Analysis of Seasoning Samples

To evaluate the flavor profiles of different seasonings, E-nose technology was used to detect the overall flavor profile of the three seasoning samples. Based on E-nose analysis (Figure 1b), the E-nose radar profiles of the three seasoning samples were basically identical, among which S3 (hydrogen), S7 (flammable gas of short-chain alkanes), S10 (sensitive to hydrogen), and S23 (sensitive to alkanes, olefins, and hydrogen) sensors showed small response values to volatile flavoring compounds of the seasoning samples. These volatile flavoring compounds had little effects on the flavor of the seasoning samples [29]. Compared with other sensors, S5, S8, S11, S13, S14, S17, S18, S21, S24, S25, S26, and S27 sensors were more responsive to volatile flavoring compounds in various seasoning samples. In particular, S18 (sensitive to alcohols, aldehydes, ketones, and aromatic compounds), S24 (sensitive to alkanes, carbon monoxide, enaldehydes, alcohols, nitrogen oxides, ketones, and aldehydes), and S25 (sensitive to aromatic hydrocarbons, aliphatic hydrocarbons, alicyclic hydrocarbons, and halogenated hydrocarbons) were the most significant sensors. The three flavoring samples might have higher contents of alcohols, aldehydes, ketones, hydrocarbons, nitrogen oxides, and aromatic compounds. In contrast, the response values of S18, S24, and S25 sensors in F were significantly (*p* < 0.05) greater than those in S. The response values of S18, S24, and S25 sensors in SF were significantly higher than those in S and F, indicating that SF may contain a higher number of alcohols, aldehydes, ketones, and aromatic compounds. This phenomenon may be attributed to the presence of *C. glutamicum* and the use of soybeans to separate sugars from protease hydrolysate to produce corresponding volatile flavoring compounds such as alcohols, aldehydes, ketones, and aromatic compounds through glycolysis, pentose phosphate, shikimate, and their overflow pathways [30].

To further analyze the differences in the flavor profiles of different seasoning samples, PCA of E-nose data was performed to obtain response signals (Figure 1c). PC1 and PC2 are two key PCA components, and the proportions of the two most important directions in the original data reflect the degree of data differences in these directions [11]. As shown in Figure 1b, PC1 (11.7%) and PC2 (80.8%) accounted for 92.5% of the data differences, indicating that these two PCs reflect the overall flavor profile of the seasoning sample. Samples S, F, and SF were distributed in different regions without any overlap, showing a significant (*p* < 0.05) difference between their PCs. Samples S, F, and SF were located close to each other, with a high flavor similarity. Therefore, the combination of E-nose and PCA could effectively distinguish the overall flavor profiles of different seasoning samples.

### 3.3. Volatile Flavoring Compounds in Seasoning Samples Detected via GC–MS

The quality of the seasoning sample is the main factor that determines its aroma and the volatile flavoring compounds produced during fermentation [31]. Under the combined action of microorganisms and enzymes, fats, proteins, and other substrates present in the SPI enzymatic hydrolysate are broken down to produce free amino acids, peptides, aldehydes, ketones, and other nutrients [16,32,33]. Small-molecule flavoring compounds, such as alcohol, pyrazines, and furan compounds, play a key role in determining the flavor and quality of the final seasoning. To more accurately understand the varieties of volatile flavoring compounds in different seasoning samples, GC–MS was used to detect the category and relative content of volatile flavoring compounds in different seasoning samples (Figure 2a,b, and Table A2). From the three seasoning samples, a total of 91 volatile flavoring compounds, including 13 alcohols, 3 aldehydes, 19 ketones, 6 esters, 8 acids, 4 amines, 10 aromatics, and 28 heterocyclic compounds, were identified. Most compounds were found in fermented seasoning products. Examples of such products include shrimp paste [34], soy sauce [35], and fish sauce [36].

Alcohol compounds are the by-products of lipid oxidation, which may impart a lipid flavor to the seasoning sample [37]. The types and abundances of various alcohols in a seasoning sample vary depending on how the sample was prepared. Among the 3 seasoning samples, 13 alcohol compounds were detected, and SF had the most types of alcohols (10), followed by F (9) and S (8). The compounds 2,6-dimethyl-3-heptanol, 4-ethylcyclohexanol, 5-indenol, (S)-(+)-3-methyl-2-butanol, and 1-[1-methyl-2-(2-propenyloxy) ethoxy]-2-propanol were detected in S, F, and SF samples and imparted a fruity flavor to the seasoning [38]. However, 1-pentanol, 1-amantanol, 1-octanol, 2-ethylhexanol, and citronellol were detected only in F and SF and were closely related to the typical mushroom odor and grassy aroma [31]. SF had a higher relative content of alcohols (20.0789%) than S and F. Straight and branched alcohols may be formed during microbial fermentation or as degradation products of lipid oxidation; however, alcohols with higher odor thresholds may have limited effects on the overall flavor of the seasoning [39].

Aldehydes with low odor thresholds and strong odor characteristics play important roles in determining the flavor profiles of seasonings. In the three seasoning samples, three aldehydes were detected, namely, 3-(4-hydroxyphenyl)-propionaldehyde, 2,4-dimethylbenzaldehyde, and hexanal. The compound 2,4-dimethylbenzaldehyde had the highest content and contributed the most to the flavor of seasonings, mainly imparting almond, cherry, and nut flavor aromas [40]. In addition, hexanal, which imparts fatty and grassy aromas, was potentially formed through various biosynthetic pathways [41]. Therefore, the relative content of *C. glutamicum* in fermented samples was higher, indicating that the samples fermented by *C. glutamicum*, particularly those fermented by SPI enzymatic solution (SF), have stronger fatty and grassy aromas.

Meanwhile, in plant-based fermentation products, ketones may be produced via amino acid metabolism or lipid oxidation. Their sensory thresholds are higher than those of aldehydes, and they have a unique fragrance and fruity aroma [42]. Nineteen ketones were detected in the three seasonings. The number of ketone types in SF (16) was significantly higher than the numbers of ketone types in S (7) and F (7), and the relative content of ketones in SF (3.2954%) was significantly higher than the relative contents in F (2.4120%) and S (2.3856%) (*p* < 0.05). The significant increase in the number of ketone types and relative content of ketones in SF may be closely related to the use of SPI enzymatic hydrolysate by *C. glutamicum* for lipid decomposition or amino acid metabolism [11,43]. Among these ketones, acetone, 2-heptanone, and 2-nononone have minty and fruity aromas; 3-octene-2-one, 2-decanone, and acetoin have fatty and dairy aromas and are detected only in SPI seasoning samples, which may be the key compounds affecting the flavor and quality of the seasonings [44].

Moreover, acids formed through the oxidative degradation of fatty acids usually have a pungent, unpleasant odor [40]. Eight types of acids were detected in the three seasoning samples, and the relative proportion of acid substances was 7.1080–8.1422%. Significant differences were noted among the seasoning samples (*p* < 0.05), mainly manifested as the contents of octanoic acid, decanoic acid, and hexanoic acid, which were detected in all seasoning samples [11,27]. Therefore, acids have little effect on the flavor of the seasoning samples.

Moreover, acids formed mainly from the oxidative degradation of fatty acids usually have a pungent unpleasant odor [45]. Eight kinds of acids were detected in the three seasoning samples, and the relative proportion of acid substances was 7.1080–8.1422%. Significant differences were found among the seasoning samples (*p* < 0.05), mainly manifested as octanoic acid, decanoic acid, and hexanoic acid, which were detected in all seasoning samples, Therefore, acids have little effect on the flavor of the seasoning samples.

Compared with S and F, the relative ester content in SF was higher (7.5797%), with the relative content of methyl 2-hydroxy-4-methylvalerate being the highest (3.4326%), followed by phenethyl acetate (3.3734%). The high content of esters may be due to the catalytic effects of esterase secreted by microorganisms on acids and alcohols during fermentation [46]. Many esters have fruity and floral flavors that could neutralize bitter and unpleasant odors caused by fatty acids and amines [47]. Meanwhile, esters may be produced via the cracking of hydrocarbons, lipids, and other compounds at high temperatures or from the aromatic substances present in spices, which were the second most prominent compounds in the seasoning samples after heterocyclic compounds. A total of 10 aromatic substances were detected. The variation trend was similar to that of other compounds, and the types and relative contents of aromatic substances in different seasonings were significantly different (*p* < 0.05). In particular, SF prepared using SPI enzymatic hydrolysate as the base material of *C. glutamicum* had higher ketone contents than S and F. The relative content of ketones was also high, which was 6.2846% in SF, 6.2267% in F, and 5.5899% in S, which may have caused the differences in the flavor of seasonings, consistent with the significant difference in the response values of E-nose S18 and S25 sensors (Figure 1a).

Heterocyclic compounds are mainly produced via the Maillard reaction and fatty oxidation, and their odor threshold was low [37]; thus, they have greater effects on the overall flavor of seasonings. In total, 28 heterocyclic compounds were detected in the three seasonings, of which 18, 22, and 23 were detected in S, F, and SF, respectively. The relative content of heterocyclic compounds in SF (42.3100%) was higher than the relative contents in S (41.9973%) and F (41.1449%). Kleekayai et al. [48] reported that the main aroma substances in traditional Thai fermentation seasonings were nitrogen-containing compounds, particularly pyrazines. Pyrazine is an important class of nitrogenous flavoring compounds that mainly imparts meaty and roasted aromas. Pyrazine accounted for 40.2509% and 53.0544% of the 22 (F) and 23 (SF) heterocyclic classes detected in fermentation samples, respectively. Tetramethylpyrazine was the main pyrazine that imparted a unique flavor, accounting for 12.5390% and 12.1259% of the relative content in F and SF, respectively, followed by 2,6-dimethylpyrazine. The relative contents were 1.2300% and 1.4220% in F and SF, respectively. In addition, various heterocyclic flavoring compounds such as furan, pyridine, pyrrole, and thiazole were detected in the seasoning samples, which have a unique flavor, such as 2,3-dihydrobenzofuran, 2-n-heptylfuran, 2-acetyl pyrrole, and 3-tert-butyl-1H-1,2,4-triazole that could impart flavoring fragrance, fragrance, lactone aroma, pork aroma, and nut aroma [49]. These results indicate that nonenzymatic browning and microbial mechanisms during fermentation could produce various heterocyclic flavoring compounds in seasonings, which was conducive to enriching the aroma richness in the final product [31].

Therefore, these results indicate that microbial fermentation can induce aroma changes in seasonings. When SPI enzymatic hydrolysate was used as the base medium to prepare new seasonings, the variety and content of volatile flavoring substances in products increased significantly, resulting in the formation of various fermentation aromas, such as cheese and meaty aromas, in SF, consistent with the results of the previous sensory analysis.

### 3.4. Correlation Analysis of Volatile Flavoring Compounds in Seasoning Samples

To visualize the relationship between different volatile flavoring substances in seasoning samples, PCA and heatmap clustering were used for analysis. The PCA (PCA-X) diagram (Figure 3a) highlights the differential characteristics of volatile flavoring substances in different seasoning samples. R2X [1] and R2X [2] explained 53.6% and 40.6% of the variances in the PCA-X plots, respectively, with a cumulative explanatory variance of 94.2%, and the confidence of the PCA reached 95.0%. Dispersion was apparent among the three seasoning samples, indicating significant differences. As shown in the biplot (Figure 3b), most of the volatile flavoring compounds had greater effects on SF, whereas a few volatile flavoring compounds influenced S and F. The relative contents of volatile flavoring compounds were color-coded in the heatmap (the higher the content of related species in the corresponding seasoning sample, the darker the red color, whereas the lower the content, the darker the blue color), and the types of flavoring compounds were horizontally clustered (Figure 3c). SF contained a higher number of volatile flavoring compounds, particularly alcohols (1-pentanol, 1-adamantanol, and benzyl alcohol), aldehydes (3-(4-hydroy-phenyl)-propionaldehyde and hexanal), ketones (acetone, 2-heptanone, 2-nonone, 3-octen-2-one, 2-cyclohexen-1-one, 3,5-dimethyl-, 3,5-diethyl-2,6-dimethylcyclohex-2-en-1-one, bicycl [3.3.1]nonane-2,6-dione, 2,3-pentanedione, 2, 3-pentadecanone, 2, 3-pentadecanone, 2-hexanedione, 2-pentadecanone, and acetoin), esters (gamma-octanoic lactone and 4-hexanolide), aromatics (olivetol, 4-hydroxy-3-methoxystyrene, benzimidazole, naphthalene, and benzothiazole), and heterocyclics (2,5-dimethyl pyrazine, 2-butyl-3-methylpyrazine, n-amylpyrazine, 2-propylpyrazine, tetramethylpyrazine, 1-(2-furyl)pentan-1-one, pyridine, 2-acetyl pyrrole, 3-tert-butyl-1H-1,2,4-triazole, dodecamethylcyclohexasiloxane and pyrazine, and 2,3,5-trimethyl-6-propyl-). Among them, naphthalene, benzothiazole, 2, 5-dimethylpyrazine, 2-valerylfuran, and tetramethylpyrazine contributed significantly to the flavor of seasoning samples. They may exert synergistic enhancement effects along with low contents of other aroma substances [50]. During the fermentation of SPI enzymatic hydrolysate, the relative contents of the abovementioned compounds increase significantly, which greatly improves the flavor of fermentation samples.

To further understand the differences in volatile flavoring compounds among different seasoning samples, an OPLS-DA regression model combined with variable importance in projection (VIP) was used to screen the differential compounds in different seasoning samples (Figure 4). The addition of speciation variables to compensate for the limitations of PCA could better reflect the interspecific differences between seasoning samples than PCA. In this model, R^2^X and R^2^Y represent the explanatory rates of the constructed model to the X and Y matrices, respectively. Q^2^ represents the predictive ability of the model, and the closer the R^2^ and Q^2^ values are to 1, the higher the degree of fit [51]. As shown in Figure 4a, the variance explanation rate of the OPLS-DA model was 94.9% (48.2% and 46.7%; Figure 4a), and the difference between the varieties of compounds was apparent (distributed in different quadrants), indicating that the model was more effective in explaining the differences in volatile flavoring compounds in the three seasoning samples. The accuracy of the OPLS-DA model was demonstrated by 200 countervalidated alignment tests (Figure 4b), where the R^2^X, R^2^Y, and Q^2^ values were 0.953, 1, and 0.991, respectively, indicating that the model had better predictive power and could better explain and predict differences between seasoning samples.

According to the results of OPLS-DA model analysis, VIP is a commonly used parameter to evaluate the contribution degree of variables. Variables with VIP > 1.0 could reflect some differences in statistical models [52]. Therefore, in the current study, we selected 24 volatile flavoring compounds with significant differences in the model and VIP > 1.0 as the index (Figure 4c), including 2-(aziridin-1-yl)ethanamine, 2,3-dihydro-2-methyl-,3,4-dimethoxycinnamic acid, tetramethylpyrazine, methyl 2-hydroxy-4-methylvalerate, (S)-(+)-3-methyl-2-butanol-, octanoic acid, phenethyl acetate, n-pentylpyrzine, 2,3-dihydrobenzofuran, hexanoic acid, 2-methyl-4-methoxyaniline, 2-acetyl-3,5-dimethylpyrazine, 4-ethylcyclohexanol, 2-ethyl-6-methyl-, pyrazine, ethyltrimethyl-(8CI,9CI), 3-heptanol, 2,6-dimethyl-, 2,4-dimethylbenzaldehyde, 2,6-dimethylpyrazin, aniline, 3-tert-butyl-1H-1,2,4-triazole, acetamide, 1-amantadanol, and capric acid. Among them, 2-acetyl-3-5dimethylpyrazine, pyrazine, ethyltrimethyl-(8CI,9CI), and 2,6-dimethylpyrazine are the main flavoring compounds produced by *C. glutamicum* using a soybean-isolated protease solution as the base material, primarily imparting nutty and chocolate aromas [12].

### 3.5. Analysis of the Relative Odor Activity Values

No direct relationship was noted between the relative content of volatile flavoring compounds and the overall flavor profile. The contribution of volatile flavoring compounds to the overall flavor is determined based on the relative content and sensory threshold [53]. Therefore, *ROAV* analysis of flavoring compounds was performed. In this method, species with *ROAV* ≥ 1 are classified as key flavoring compounds, and those with 0.1 ≤ *ROAV* < 1 have an important modification effect on the overall flavor. That is, the higher the *ROAV*, the greater the contribution of the compound to the overall flavor. Only flavoring compounds with *ROAV* ≥ 0.1 were selected in this study [54].

As shown in Table 1, 32 key flavoring compounds (12, 19, and 22 compounds from S, F, and SF, respectively) and 7 modified flavoring compounds were identified from the three seasonings, mainly including heterocyclic substances, alcohols, ketones, esters, and aromatic substances. Eight of the key flavoring compounds were present in three different seasonings, including gamma-octanoic lactone, octanoic acid, decanoic acid, benzothiazole, 2,5-dimethylpyrazine, n-amylpyrazine, tetramethylpyrazine, and pyridine. These flavoring compounds were relatively abundant in each seasoning sample. In addition to the high flavor thresholds of caprylic acid (3.00 mg/kg) and n-capric acid (10.00 mg/kg), the other key compounds greatly contribute to the flavor of the seasoning because of their low flavor thresholds. For example, tetramethylpyrazine was a major contributor to the flavor profile of the seasoning, with the highest *ROAV* (100) and a low odor threshold (2.53 mg/kg) to impart flavoring with nutty, peanut, and coffee aromas [42]. Gamma-octanoic lactone can impart a sweet herb taste to the seasoning and is a key ester in reducing the spicy taste of fatty acids and the bitterness of amino acids (*ROAV* 22.47–51.01) [55]. Benzothiazole is a key aromatic compound with an extremely low flavor threshold (0.08 mg/kg), imparting a nutty flavor to the seasoning. Furthermore, 2,5-dimethylpyrazine and n-amylpyrazine are pyrazine compounds, which are the products of the Maillard reaction after fatty oxidation and mainly impart a meaty and roasted aroma [56]. These results indicate that the key flavoring compounds of S, F, and SF have high similarity; thus, the flavors of the three seasonings are similar. This may explain why the three seasonings in Figure 1a are located relatively close to each other.

In addition, six key flavoring compounds, including citronellol, 3,5-octadien-2-one, 2-ethyl-3, 3-ethyl-2,5-dimethylpyrazine, 2-ethyl-5-methylpyrazine, 2,3-dimethyl-5-ethylpyrazine, and pyrazine, 2,5-dimethyl-3-propyl-(8CI,9CI) were detected in S. Compared with S and F, the types and contents of key flavoring compounds in SF were significantly different, showing that 1-octanol, acetone, 2-heptanone, 2-nonone, 3-octen-2-one, 4-hydroxy-3-methoxystyrene, naphthalene, 2-butyl-3-methylpyrazine, and 2-propylpyrazine were only detected in SF. Therefore, the difference in the number and content of SF key flavoring compounds in the samples is an important factor affecting the difference in flavor profiles. In addition, the *ROAV* values of benzyl alcohol, 2-ethylhexanol, 2-phenylethyl propionate, gamma-octanoic lactone, 2-methylpyrazine, and 2-acetyl pyrrole were all <1, which had flower, fruity sweetness, baking, and chocolate aromas and played an important role in the overall flavor of the seasoning. The differences in the content and types of these important and modified flavoring compounds contribute to the unique flavors of different seasonings.

### 3.6. Correlation Analysis between E-Nose Data and Important Volatile Flavoring Compounds

PCA-X was used to analyze the correlation between flavor profiles and characteristic flavoring compounds. Based on the 24 volatile flavoring compounds with VIP > 1.0 (Figure 4c), combined with *ROAV* values (Table 1), 8 (Figure 5a) important flavoring compounds (*ROAV* ≥ 0.1) and 27 E-nose responses (Table A1) were selected to construct the loading plot of the PCA-X model, also known as the correlation plot. The coordinates of each variable in Figure 5b correspond to the correlation and directivity of PC1 and PC2, respectively. Variances of 59.5% (PC1) and 15.8% (PC2) were noted, and the cumulative contribution rate was 75.3%, indicating that PC can better reflect the overall information of the sample and can be effectively used in the correlation analysis between flavor profiles and characteristic flavoring compounds. The shorter the distance between the response of the compound and E-nose data, the higher the correlation between the compound and sensor response. Among them, S2, S6, S9, S11, S15, S19, S22, S24, S25, and S26 were distributed in the square direction of PC1 and PC2 and were close to 1-amantadanol, octanoic acid, hexanoic acid, and n-amylpyrazine, indicating that these response signals are positively correlated with the contents of these important compounds. S1, S5, S8, S10, S13, S14, S17, S18, S21, and S27 were distributed in the positive direction of PC1 and the negative direction of PC2 and clustered together with 2,6-dimethylpyrazine and tetramethylpyrazine, indicating that these sensors were positively and strongly correlated with 2,6-dimethylpyrazine and tetramethylpyrazine. The results revealed that PCA-X could analyze the positive and negative correlations between the E-nose response signal and volatile flavoring compounds. Therefore, E-nose can rapidly identify the unique flavor of different flavoring samples through the specific response to volatile flavoring compounds.

## 4. Conclusions

High-quality flavor is a critical factor in attracting consumers to products, thus making research and innovation in flavor development a focal point within the food industry. In this study, a new method to improve the flavor of traditional seasoning was proposed, wherein SPI enzymatic hydrolysate was used as the fermentation base of *C.glutamicum*. Based on sensory evaluation, E-nose, and GC–MS methods, the overall flavor and volatile flavoring compounds of different seasoning samples were analyzed. Compared with S and F, SF had a richer aroma and roasted flavor. Its response signals to E-nose sensors S18 (aromatic compounds), S24 (alcohols, aldehydes), and S25 (esters, ketones) were more significant (*p* < 0.05). A total of 91 volatile flavoring compounds, including 13 alcohols, 3 aldehydes, 19 ketones, 6 esters, 8 acids, 4 amines, 10 aromatics, and 28 heterocyclic compounds, were detected via GC–MS. Compared with S and F, SF contained a higher number of alcohols, ketones, aromas, and heterocyclic compounds, which have a stronger cheesy, fatty, and roasted odor. Furthermore, *ROAV* analysis identified 32 key flavoring compounds (*ROAV* ≥ 1) and 7 modified flavoring compounds (0.1 ≤ *ROAV* < 1). Compared with S and F, nine additional key flavoring compounds were detected in SF. Among them, pyrazines such as n-pentylpyrzine, 2,6-dimethylpyrazine, and tetramethylpyrazine were the key flavoring compounds in SF seasoning (*ROAV* ≥ 1), which could impart unique meaty and roasted aromas to SPI seasoning. In addition, PCA showed a high correlation between the E-nose response signals and key volatile flavoring compounds. These results showed that the fermentation of *C. glutamicum* can effectively increase the content and types of flavoring compounds and promote the accumulation of flavoring compounds with fragrance. Furthermore, a previous study showed that the enhancement of umami substances is closely related to *C. glutamicum* fermentation [11]. However, the impact of SPI enzymatic hydrolysate is not only related to umami and flavor but also includes nutritional and functional properties; therefore, further research on the potential nutritional and functional properties of SPI enzymolysis using *C. glutamicum* is the focus of future research. In addition, it is extremely important to determine the influence of microbial metabolism on the types and contents of flavor and umami substances produced in the fermentation enzymatic hydrolysate. Therefore, it is particularly important to understand the mechanism of microbial metabolism to produce flavor and umami substances. 

## Figures and Tables

**Figure 1 foods-13-02591-f001:**
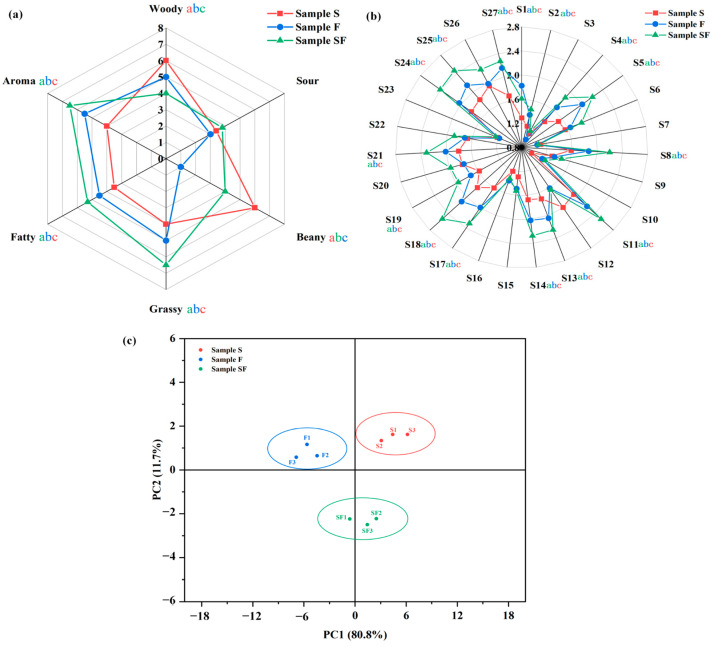
Sensory evaluation and electronic nose analysis of seasoning samples (different letters indicate significant differences [*p* < 0.05]). (**a**) Sensory scoring radar map, (**b**) odor radar map of the response of the electronic nose sensor, and (**c**) correlation of odor with the electronic nose sensor response data (PC1 and PC2 were the directions with the largest and smallest differences, respectively).

**Figure 2 foods-13-02591-f002:**
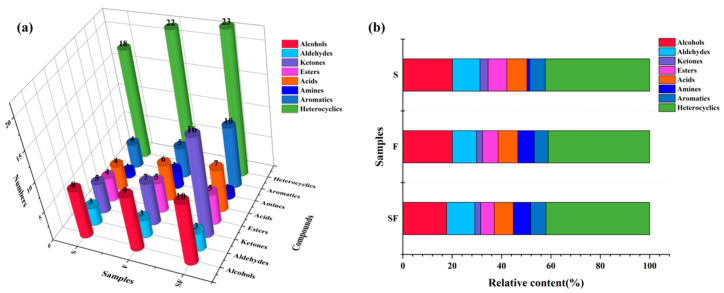
(**a**) Types of volatile compounds in three seasoning samples. (**b**) Relative levels of volatile compounds in three seasoning samples.

**Figure 3 foods-13-02591-f003:**
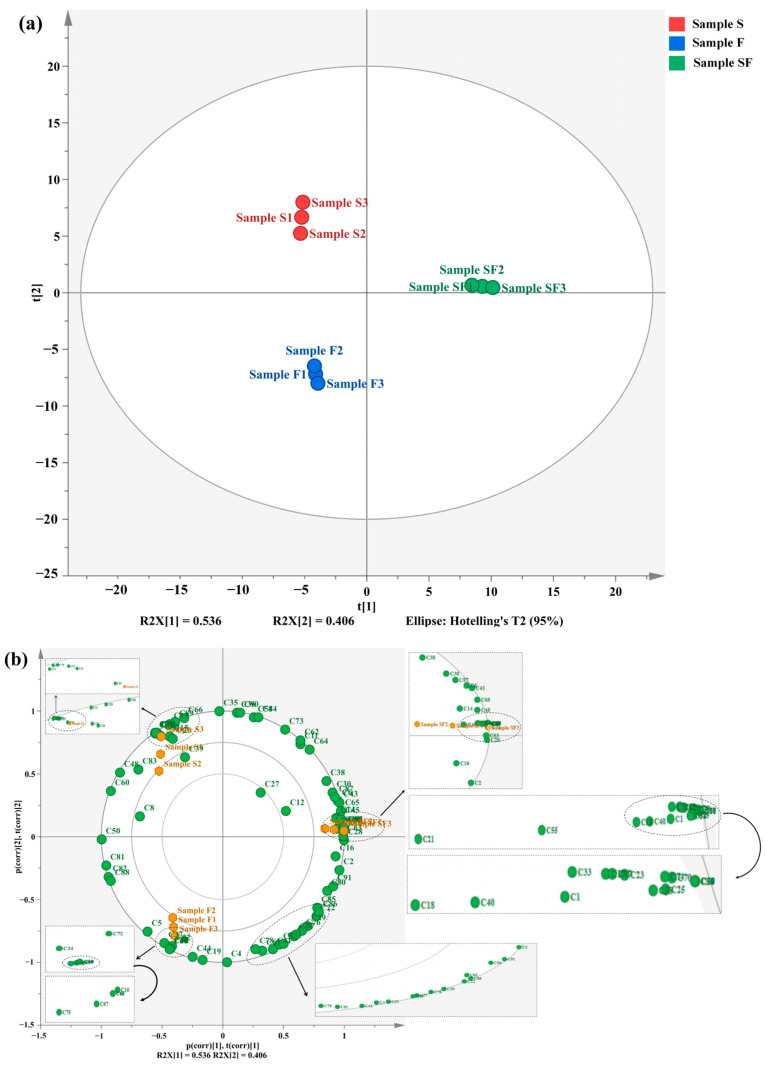
(**a**) PCA-X diagram of different flavoring samples (red, blue, and green dots represent samples S, F, and SF, respectively). (**b**) Biplots of different flavor samples (green and orange dots represent volatile flavor substances and flavor samples, respectively). (**c**) Heatmap based on the relative content distribution of volatile flavor substances in different flavor samples (red and blue colors represent the high and low concentrations of volatile flavor substances, respectively; the higher the concentration of volatile flavor substances, the darker the color).

**Figure 4 foods-13-02591-f004:**
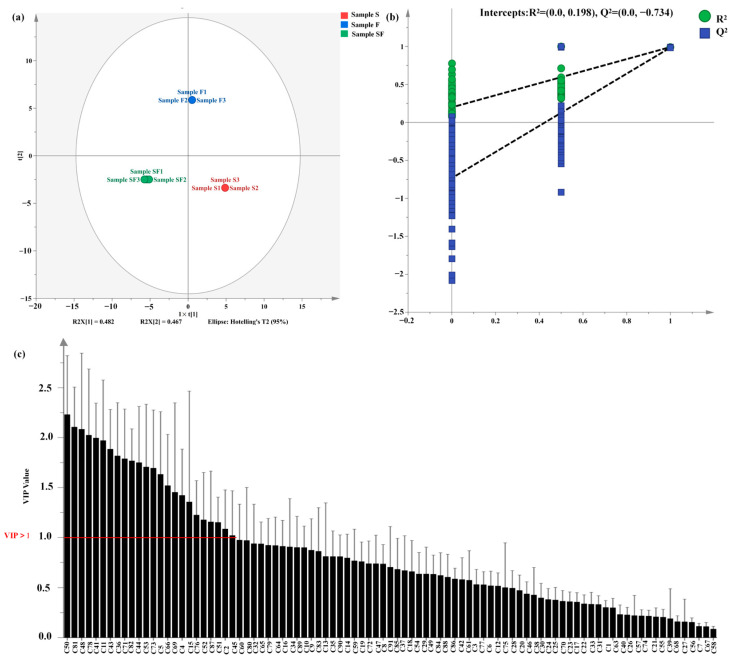
(**a**) OPLS-DA regression projection of volatile compounds. (**b**) Validation via the permutation test. (**c**) VIP value calculated based on the OPLS-DA regression model.

**Figure 5 foods-13-02591-f005:**
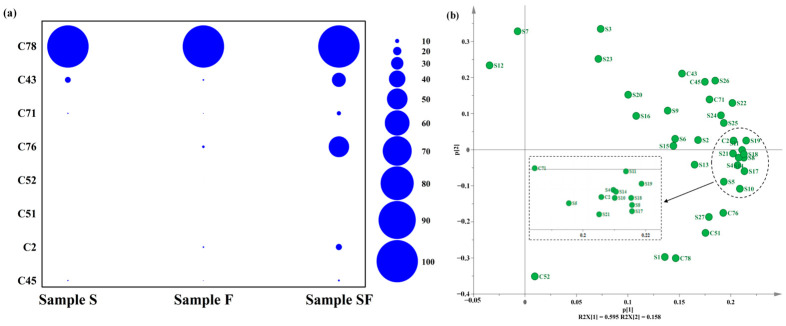
(**a**) *ROAV* values of eight important volatile compounds. (**b**) Correlation graph between the electronic nose sensor and eight important volatile flavoring compounds.

**Table 1 foods-13-02591-t001:** Relative odor activity values of volatile compounds in different seasoning samples.

No.	Compounds	CAS	Threshold Value (mg/kg)	*ROAV* Values		
				S	F	SF
**Alcohols (** **7** **)**						
C1	1-Pentanol	71-41-0	0.1502	—	3.73	13.14
C2	1-Adamantanol	768-95-6	2.0000	—	39.62	14.80
C3	1-Octanol	111-87-5	0.1258	—		46.77
C6	Benzyl alcohol	100-51-6	5.1000	0.34	—	1.28
C7	(R,R)-2,3-Butanediol	24347-58-8	0.0951	3.73	—	—
C9	2-Ethylhexanol	104-76-7	25.4822	—	0.49	—
C10	Citronellol	106-22-9	2.2000	—	5.64	—
**Ketones (6)**						
C17	Acetone	67-64-1	0.8320	—	—	3.04
C18	2-Heptanone	110-43-0	0.1400	—	—	63.91
C20	2-Nonanone	821-55-6	0.0820	—	—	51.42
C21	3-Octen-2-one	1669-44-9	0.2500	—	—	33.82
C31	Acetophenone	98-86-2	1.6000	1.57		
C34	3,5-Octadien-2-one	30086-02-3	0.1500	—	88.21	—
**Esters (** **4** **)**						
C37	Acetic acid, phenethyl acetate	103-45-7	0.2496	3.75	30.12	—
C38	Gamma-octanoic lactone	104-50-7	0.2000	48.10	22.47	51.01
C39	2-Phenylethyl propionate	122-70-3	18.0000	0.62	0.33	0.44
C40	4-Hexanolide	695-06-7	12.5000	—	—	0.09
**Acids (4)**						
C43	Octanoic acid	124-07-2	3.0000	14.08	3.94	33.66
C45	Decanoic acid	334-48-5	10.0000	2.81	1.56	4.49
**Amines (** **2** **)**						
C51	Acetamide	60-35-5	160.0000	—	0.14	0.18
C52	Aniline	62-53-3	70.1000	—	0.32	—
**Aromatics (** **3** **)**						
C56	4-Hydroxy-3-methoxystyrene	7786-61-0	0.0120	—	—	44.04
C58	Naphthalene	91-20-3	0.0060	—	—	27.71
C63	Benzothiazole	90-05-1	0.0800	3.16	2.81	18.14
**Heterocyclics (** **15** **)**						
C64	2-Methylpyrazine	109-08-0	30.0000	0.79	0.14	0.87
C65	2,5-Dimethylpyrazine	123-32-0	1.7500	6.88	2.80	26.98
C67	2-Ethyl-5-methylpyrazine-	13360-64-0	0.0160	—	12.79	—
C68	3-Ethyl-2,5-dimethylpyrazine	13360-65-1	0.0086	—	49.05	—
C70	2-Butyl-3-methylpyrazine	15987-00-5	0.2600	—	—	11.00
C71	N-pentylpyrzine	6303-75-9	9.0000	2.87	1.86	10.20
C72	5-Ethyl-2,3-dimethylpyrazine	15707-34-3	0.5300	—	16.78	—
C74	2-Propylpyrazine	18138-03-9	0.1596	—	—	6.46
C75	Pyrazine, 2,5-dimethyl-3-propyl- (8CI,9CI)	18433-97-1	2.0000	—	2.00	—
C76	2,6-Dimethylpyrazine	108-50-9	0.7180	—	5.97	49.38
C77	Pyrazine, 2,3-dimethyl	5910-89-4	0.8000	—	5.42	7.73
C78	Tetramethylpyrazine	1124-11-4	2.5250	100.00	100.00	100.00
C79	2-n-Butyl furan	4466-24-4	10.0000	2.31	—	—
C85	Pyridine	110-86-1	2.0000	12.08	10.22	15.66
C86	2-Acetyl pyrrole	1072-83-9	58.5853	—	0.07	0.145

“—” indicates that *ROAV* cannot be calculated without retrieving the relevant data of the substance.

## Data Availability

The original contributions presented in the study are included in the article, further inquiries can be directed to the corresponding authors.

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
