# Peer review of "Effect of *Corynebacterium glutamicum* Fermentation on the Volatile Flavors of the Enzymatic Hydrolysate of Soybean Protein Isolate"

_foods, 2024, doi:10.3390/foods13162591_

Round 1

Reviewer 1 Report

Comments and Suggestions for Authors

General/major comments

In this paper, the authors present a study aiming to improve the flavor characteristics of a seasoning obtained by microbial fermentation of soybean protein isolate (SPI) enzymatic hydrolysate. They compare the aromatic compositions of three samples (sample prepared by traditional enzymatic hydrolysis, sample obtained by traditional fermentation with C. glutamicum and fermentation using C. glutamicum of an enzymatic hydrolyzate of a soybean protein isolate). Overall, this is an interesting study that provides a lot of data on the aromatic profiles of the three samples tested. The authors were able to highlight a large number of aromatic molecules belonging to different chemical families and showed that the sample corresponding to the fermentation of an enzymatic hydrolyzate had the greatest aromatic richness.

The results and discussion section does not call for any particular remarks, the authors compare the three samples in terms of sensory evaluation, E-nose analysis, GC-MS analysis. Even if this remains quite descriptive, this work provides arguments in favor of fermentation of SPI enzymatic hydrolysate in order to develop new seasoning products.

Specific comments

L58: Please the new name of Lactobacillus plantarum (i.e.: Lactiplantibacillus plantarum, Zheng et al., 2020).

L141: “…on the seasoning samples…” instead of “…on he seasoning samples…”

L243: replace “which” by “whose”

Figure 3a et 3b are not fully readable

Throughout the manuscript (in the results section), the authors provide data with 4 digits after the decimal point, is it significant ?

Comments on the Quality of English Language

Proofreading by an English speaker is desirable to correct some language imperfections.

Reviewer 2 Report

Comments and Suggestions for Authors

The work is interesting.

A similar work by the same authors exist in the literature and should be used extensively in the discussion of the present study.

Line 460 and elsewhere. Why 3.00000? and not just 3?

Lines 161-167. Please explain. The headspace was used for the analysis?

Section 2.5. No internal standard? No quantification? Only qualitative analysis?

Lines 185-187. “the compound making the greatest contribution to the odor of the seasoning sample was defined as ROAVstan”. How the compound making the greatest contribution was selected since no quantitation took place?

Please add more details in sections 2.5 and 2.6.

A chemical analysis of the final products after fermentation should be added in order to provide readers details of the products.

GC/MS analysis of the unfermented SPI is needed in order to better correlate the results and to evaluate the effect of fermentation.

Line 521. “a new process was developed”??? There is no new process, just one fermentation.

Lines 537-539. Any analyses? How the authors wrote something like this?

More discussion is needed based on the previous similar works of the authors 11 and 16. Especially 11 is exactly the same work but with different analyses. The authors should discuss the results of the present study based on the analyses of the previous study 11.

Comments on the Quality of English Language

English are difficult to follow.

Please extensive revise and improve.

Some characteristic examples:

Lines 20-22. Improve this sentence.

Line 20-24. In this part the authors write the same thing in three different ways please revise.

Line 43. Please correct flavor

Line 53. “good” Please improve English.

Line 64. “bad” Please improve.

Round 2

Reviewer 2 Report

Comments and Suggestions for Authors

The manuscript has been improved after revision.

Some minor additional comments/suggestions

Line 57. Which Saccharomyces? Please add.

Line 160. “homogenized” Please provide details. Equipment etc.

GC/MS. In order to clarify. After extraction a direct injection made. Is that correct? Or from the headspace?

Figure 1. In PCA figure, it would be better to reduce the scale off axes, in order to become easier to read. A possible suggestion would be PC1 -20 to 20 and PCA2 -6 to 6.

Table 1. I would like to suggest the reduction of decimals in ROAV values to 1 or 2.

Author Response

Dear Reviewer:

Thank you very much for your careful review. Per your suggestion, We have further refined our paper in the last revision. We have made corresponding changes in the main text, which are marked in red. Our point-by-point responses to your comments are provided in this letter.

Comments 1: Line 57. Which Saccharomyces? Please add.

Response 1: Thank you for indicating this issue. Per your suggestion, we have revised “Saccharomyces” to “Saccharomyces cerevisiae” with new references (Page 2, lines 60).

Comments 2: Line 160. “homogenized” Please provide details. Equipment etc.

Response 2: Thank you for pointing this out. We have revised the text from “The mixture was homogenized for 1 min at 1,300 rpm…” to “With a magnetic stirrer (Gongyi Yuhua instrument Co., Ltd., China), the mixture is thoroughly mixed for 1min at a speed of 1,300 rpm…” (Page 4, line 168-170).

Comments 3: GC/MS. In order to clarify. After extraction a direct injection made. Is that correct? Or from the headspace?

Response 3: Thank you for raising this question. We agree that the relevant text was not properly expressed. We have revised the text accordingly (Page 4, line 171-177).

Comments 4: Figure 1. In PCA figure, it would be better to reduce the scale off axes, in order to become easier to read. A possible suggestion would be PC1 -20 to 20 and PCA2 -6 to 6.

Response 4: Thank you for your valuable comment. We have revised the coordinates in Figure 1. (Page 7, lines 278).

Comments 5: Table 1. I would like to suggest the reduction of decimals in ROAV values to 1 or 2.

Response 5: Thank you for this suggestion. Per your suggestion. We have reduced the decimal number of the ROAV values in Table 1 to 2 places(Page 14, lines 523).

Yours sincerely,

Jun Dai, Ph.D.

School of Life Sciences and Health, Hubei University of Technology, No. 28, Nanli Road, Hongshan District, Wuhan, Hubei, 430068, P.R. China

Tel: +8613476179588;

Email address: jundai@hbut.edu.cn